# Reduced Expression of Cerebral Metabotropic Glutamate Receptor Subtype 5 in Men with Fragile X Syndrome

**DOI:** 10.3390/brainsci10120899

**Published:** 2020-11-24

**Authors:** James R. Brašić, Ayon Nandi, David S. Russell, Danna Jennings, Olivier Barret, Anil Mathur, Keith Slifer, Thomas Sedlak, Samuel D. Martin, Zabecca Brinson, Pankhuri Vyas, John P. Seibyl, Elizabeth M. Berry-Kravis, Dean F. Wong, Dejan B. Budimirovic

**Affiliations:** 1Section of High Resolution Brain Positron Emission Tomography Imaging, Division of Nuclear Medicine and Molecular Imaging, The Russell H. Morgan Department of Radiology and Radiological Science, The Johns Hopkins University School of Medicine, Baltimore, MD 21287, USA; anandi1@jhmi.edu (A.N.); amathur4@jhmi.edu (A.M.); tsedlak@jhmi.edu (T.S.); smart149@jhu.edu (S.D.M.); zabecca.brinson@gmail.com (Z.B.); pankhuri.v@gmail.com (P.V.); dfwong@wustl.edu (D.F.W.); 2Clinical Research, Institute for Neurodegenerative Disorders, New Haven, CT 06510, USA; drussell@invicro.com (D.S.R.); Jennings@dnli.com (D.J.); olivier.barret@cea.fr (O.B.); jseibyl@invicro.com (J.P.S.); 3Research Clinic, Invicro LLC, New Haven, CT 06510, USA; 4Denali Therapeutics, Inc., South San Francisco, CA 94080, USA; 5Department of Psychiatry and Behavioral Sciences-Child Psychiatry, The Johns Hopkins University School of Medicine, Baltimore, MD 21205, USA; slifer@kennedykrieger.org; 6Department of Behavioral Psychology, Kennedy Krieger Institute, Baltimore, MD 21205, USA; 7Department of Psychiatry and Behavioral Sciences-General Psychiatry, The Johns Hopkins University School of Medicine, Baltimore, MD 21205, USA; 8Department of Neuroscience, Zanvyl Krieger School of Arts and Sciences, The Johns Hopkins University, Baltimore, MD 21218, USA; 9Departments of Pediatrics, Neurological Sciences, and Biochemistry, Rush University Medical Center, Chicago, IL 60612, USA; Elizabeth_Berry-Kravis@rush.edu; 10Precision Radio-Theranostics Translational Laboratories, Mallinckrodt Institute of Radiology, School of Medicine, Washington University, Saint Louis, MO 63110, USA; 11Departments of Psychiatry and Neurogenetics, Kennedy Krieger Institute, Baltimore, MD 21205, USA

**Keywords:** binding potential, caudate nucleus, *FMR1* gene, Fragile X Mental Retardation Protein (FMRP), genetic mutation, magnetic resonance imaging (MRI), mosaicism, neuropsychological testing, positron emission tomography (PET), 3-[^18^F]fluoro-5-(2-pyridinylethynyl)benzonitrile ([^18^F]FPEB)

## Abstract

Glutamatergic receptor expression is mostly unknown in adults with fragile X syndrome (FXS). Favorable behavioral effects of negative allosteric modulators (NAMs) of the metabotropic glutamate receptor subtype 5 (mGluR_5_) in *fmr1* knockout (KO) mouse models have not been confirmed in humans with FXS. Measurement of cerebral mGluR_5_ expression in humans with FXS exposed to NAMs might help in that effort. We used positron emission tomography (PET) to measure the mGluR_5_ density as a proxy of mGluR_5_ expression in cortical and subcortical brain regions to confirm target engagement of NAMs for mGluR_5_s. The density and the distribution of mGluR_5_ were measured in two independent samples of men with FXS (*N* = 9) and typical development (TD) (*N* = 8). We showed the feasibility of this complex study including MRI and PET, meaning that this challenging protocol can be accomplished in men with FXS with an adequate preparation. Analysis of variance of estimated mGluR_5_ expression showed that mGluR_5_ expression was significantly reduced in cortical and subcortical regions of men with FXS in contrast to age-matched men with TD.

## 1. Introduction

### 1.1. Background

Fragile X syndrome (FXS) is caused by expansion full mutation (≥200 CGGs) of the fragile X mental retardation 1 *(FMR1*) gene leading to epigenetic silencing of the gene, resulting in reduction of its product: fragile X mental retardation protein (FMRP) [1]. FXS is the leading single-gene cause of inherited intellectual disability (ID) and autism spectrum disorder (ASD) [2,3]. Indeed, studies of humans with FXS have consistently demonstrated a wide range of global neurobehavioral impairments [4,5,6,7,8,9]. This is not surprising, as FMRP controls translation around 4% of mRNA in human brains. To illustrate, FMRP binds brain mRNAs, inhibits synthesis of a myriad of proteins [10], and increases the dosages of FMRP target proteins (over 600 to date) of relevance to ASD [11]. The FMRP expression in the brain is the ultimate factor determining the severity of the neurobehavioral phenotype [12]. The absence of adequate FMRP results in overactive glutamatergic signaling of group 1 metabotropic (mGluR_1_ and mGluR_5_) pathways, and consequently overactive downstream signaling cascades, such as the mammalian target of rapamycin (mTOR) and mitogen-activated protein kinase (MAPK)/extracellular signal-regulated kinase (ERK). The overactive downstream signaling leads to excessive protein synthesis in an *fmr1* knockout (KO) mouse model [13]. Namely, abnormal mGluR_5_-modulated long-term depression (LTD) in the hippocampus in the *fmr1* KO model led to “the mGluR_5_ theory” of neuronal dysfunction in FXS [14]. Indeed, the abnormal signaling in the absence of FMRP is associated with aberrant synaptic plasticity and immature dendritic spine morphology. The abnormal excitation–inhibition that leads to an excessive de novo protein synthesis also occurs in humans with FXS [15,16,17,18]. Targeted treatment studies using mGluR_5_ negative allosteric modulators (NAMs) then unfolded in both the *fmr1* KO mouse model and in humans with FXS [3,12]. Yet, mGluR_5_ expression in animal studies and in autopsy studies of humans with FXS produced inconsistent results. Moreover, mGluR_5_ expression in vivo has not been measured in humans with FXS.

Although a necropsy study pooling human brains with FXS and premutation of the *FMR1* gene (PM, 55–200 CGGs) showed increased mGluR_5_s and marginal protein overexpression [19], these studies do not exist in the living human brain. Since the limited necropsy findings may represent the changes in agonal and post-mortem periods, in vivo measurement of the expression of mGluR_5_s is needed, which may bring an initial insight into failed clinical trials that used investigational agents acting on mGluR_5_ in humans FXS. Novel, specific mGluR_5_ ligands that allow quantitative measurement of the density and distribution of mGluR_5_s in the brain, such as 3-[^18^F]fluoro-5-(2-pyridinylethynyl)benzonitrile ([^18^F]FPEB), require studies of feasibility.

Indeed, quantification of mGluR_5_ expression in the living human brain of men with FXS is needed to help understand results of past mGluR_5_ trials in humans with FXS, and to help provide information for successful clinical trial designs. For example, an alteration of expression of mGluR_5_s, such as internalization of membrane mGluR_5_s, may be one possible explanation for the negligible therapeutic effect of NAMs in “failed” clinical trials of humans with FXS [20]. Since proteins and receptors occupy different locations on the membranes, the living brain may show protein overexpression and reduction of receptors due to receptor internalization or other alterations. Thus, the use of [^18^F]FPEB may serve as an effective tool to confirm target engagement of NAMs for mGluR_5_s.

### 1.2. Measurement of mGluR_5_s in the Living Human Brain

While several techniques exist to estimate the concentration of glutamate in the living brain, including magnetic resonance imaging (MRI) and brain biopsy, positron emission tomography (PET) uniquely provides the optimal means to measure mGluR_5_s. For these reasons, radiotracers that bind to mGluR_5_ in the living brain and can be visualized with PET are promising tools to quantify the density and the distribution of mGluR_5_s in humans with FXS.

#### [^18^F]FPEB

3-[^18^F]fluoro-5-(2-pyridinylethynyl)benzonitrile ([^18^F]FPEB), a novel, specific mGluR_5_ ligand to quantitatively measure the density and distribution of mGluR_5_s in the brain regions of humans with FXS through PET [21,22,23,24,25,26,27,28,29,30] (Figure 1), constitutes an effective tool to confirm target engagement of mGluR_5_s of relevance to clinical trials of NAMs for individuals with FXS [31].

Specifically, [^18^F]FPEB [22,34,35,36,37] has been shown to demonstrate high uptake and specific binding during the test–retest paradigm for mGluR_5_ in the anterior cingulate gyrus, putamen, caudate nucleus, and frontal, parietal, and temporal cortices [28,29]. [^18^F]FPEB has demonstrated deficits in the striatal and neocortical mGluR_5_s in people with mild Huntington’s disease [34,35] and increments in the mGluR_5_s in people with mild Parkinson’s disease [36,37,38] and men with ASD [25].

We sought to quantify the density and distribution of mGluR_5_ expression in FXS [39,40,41,42,43,44,45,46,47,48,49,50,51] by means of PET.

## 2. Materials and Methods

### 2.1. Participants

#### 2.1.1. Recruiting Sites

The study is approved by Johns Hopkins Medicine Institutional Review Board IRB 169 249. The protocols for the study of humans with FXS were approved by the Institutional Review Boards of the Institute for Neurodegenerative Disorders (IND) in New Haven, Connecticut [52] and Johns Hopkins University (JHU) in Baltimore, Maryland [53,54]. Since exposure to radioactivity in PET constitutes greater than minimal risk, this pilot study was restricted to adults. Written informed consent was obtained from each participant at both locations.

#### 2.1.2. Inclusion Criteria

Inclusion criteria for all subjects were age 18–60 years and a diagnosis of FXS based on *FMR1* DNA gene testing by PCR/Southern Blot, supplemented by clinical neurobehavioral profiling [52].

#### 2.1.3. Exclusion Criteria

Exclusion criteria were clinically significant abnormal laboratory values and/or clinically significant unstable serious medical, neurological, or psychiatric illnesses [52].

#### 2.1.4. Institute for Neurodegenerative Disorders (IND)

Participants with FXS had completed genetic and other evaluations before traveling to the IND with a caregiver. One day after arrival to the IND, they underwent a screening assessment to confirm the inclusion and exclusion criteria, neuropsychological evaluation, mock scanner training, and PET scans. Participants with TD were recruited from community residents.

Seven men with FXS (mean age 25 ± 5, range 23–34 years) recruited from Rush University Medical Center, Chicago, Illinois, and three age-matched men with typical development (TD) (mean aged 32 ± 4, range 27–39 years) participated in the protocol. Clinical and demographic data [55] confirmed that all participants met the criteria to receive the adult dose of 185 megabecquerels (MBqs) (5 millicuries (mCis)) of [^18^F]FPEB.

#### 2.1.5. Johns Hopkins University (JHU)

Four men with FXS (mean age 28 ± 9, range 19–41 years) were recruited from the Kennedy Krieger Institute, Baltimore, Maryland, and Rush University Medical Center, including referrals from the Fragile X Online Registry With Accessible Research Database (FORWARD) of the National Fragile X Foundation (NFXF) funded by the Centers for Disease Control and Prevention (CDC), Atlanta, Georgia. The results of two of the four men with FXS (mean age 25.5 ± 2.1, range 24–27) who completed PET scans were reported in this article. Findings were compared and contrasted with five age-matched historical control men with TD who had already completed similar protocols (mean age 29.6 ± 6.02, range 24–39 years) [25,30]. Clinical and demographic data confirmed that all participants met the criteria to receive the adult dose of 185 MBqs (5 mCis) of [^18^F]FPEB [55].

### 2.2. Assessments

#### 2.2.1. Institute for Neurodegenerative Disorders (IND)

Assessments of participants with FXS (600 to 1600 CGGs) included mean FMRP of 0.047 ± 0.04 ng/microgram total protein (reference mean FMRP of 0.87 for healthy normal controls with TD), reading level under first grade level, and scores for the Dementia Screening Questionnaire for Individuals with Intellectual Disabilities (DSQIID) [56] ranging from 0 to 2 [55].

#### 2.2.2. Johns Hopkins University (JHU)

Assessments of participants with FXS (>200 CGGs) included mean FMRP of 0.00025, mean abbreviated IQ [57,58] of 48.5 ± 2.12 [55], and mean Adaptive Behavior Composite Standard Score [59] of 71.5 ± 26.16 [55].

### 2.3. Procedures

#### 2.3.1. Magnetic Resonance Imaging (MRI)

##### IND

In order to minimize anxiety and claustrophobia, participants with FXS at the IND did not undergo MRI. Participants with TD underwent MRI to compare and contrast with other cohorts (*N* = 3) [52,55].

##### JHU

Participants with FXS and TD at JHU underwent MRI (Table 1) to rule out intracranial pathology and to co-register with PET [55].

#### 2.3.2. Positron Emission Tomography (PET)

##### IND

With the head stabilized by a gauze strip taped across the forehead and a rounded head holder, each participant received an intravenous bolus injection of 185 MBqs (5 mCis) of [^18^F]FPEB [30] at 1 PM, followed by scans on an ECAT EXACT HR+ PET attaining an axial resolution of approaching y = 4–5 mm [61], with consecutive 6 × 300 s frames performed for 90 to 120 min after the injection time.

Statistical parametric mapping (SPM) [33] was applied to PET frames to obtain regional time (radioactivity) curves (TACs). The ratio of uptake in the volumes of interest (VOIs) to the uptake in the whole cerebellum, a reference region with minimal [^18^F]FPEB uptake [28,29], was calculated.

##### JHU

MRI was performed an hour before PET. Each participant with FXS underwent training using a mock scanner [62,63,64]. Each participant had a custom fitted face mask made by nuclear medical technologists to hold the head in the same position throughout the scan [65,66]. After receiving intravenous bolus injections of 185 MBq (5 mCis) of [^18^F]FPEB (30), participants underwent PET scans on a high resolution research tomograph (HRRT), attaining an axial resolution approaching 2.3 to 2.5 mm [67,68] at 1 PM for 90 min.

VOIs were obtained automatically of cortical regions with Freesurfer 6.0 [69,70] and of subcortical regions with the subcortical segmentation tools of the software library of the Oxford Centre for fMRI of the Brain [71,72,73]. VOIs were transferred from MRI to PET space according to MRI-to-PET co-registration parameters obtained with the co-registration module [74,75] of statistical parametric mapping (SPM) [33] and applied to PET frames to obtain regional TACs. With the cerebellar white matter as the reference VOI [28,29], regional BP_ND_s [32] were obtained by reference tissue graphical analysis (RTGA) [76,77].

#### 2.3.3. Comparisons and Contrasts of Cohorts from the IND and JHU

In order to directly compare and contrast data from both cohorts including nine men with FXS (mean age 27.21 ± 4.17, range 22.3–33.6) and eight historical control age-matched men with TD who had already completed similar protocols (mean age 30.63 ± 5.58, range 24–39 years) [25,30,52,55], we approximated the data with several estimates by means of multiple assumptions: (1) consistency over time of both standard uptake volume ratios (SUVRs) and distribution volume ratios (DVRs), since JHU PET scans spanned 0 to 90 min after radiotracer injections, while IND PET scans spanned 90–120 min; (2) approximately equivalent anatomical brain regions, as MRI-based segmentation was utilized for VOI analysis at JHU, but an atlas-based approach was applied to the IND data; and (3) approximately equivalent analyses, although the resolution of the scans from the IND was approximately twice the resolution of scans from JHU. Using the measurements of SUVR from the IND dataset, we derived estimates of binding potentials as DVR-1 [78], which were pooled with the comparable BP_ND_ estimates from the JHU data.

## 3. Results

### 3.1. mGluR_5_s in Humans with FXS

#### 3.1.1. IND

The density of mGluR5s was comparable in the men with FXS and the men with TD (Figure 2) [55].

#### 3.1.2. JHU

Participant JHUFXS1 withdrew before scans due to a family emergency.

Participant JHUFXS2 completed both MRI and PET scans in one day without mock scanner training.

Due to scheduling problems, MRI and PET scans were conducted on Participant JHUFXS3 on separate days a week apart without mock scanner training. Despite the administration of 2.0 mg of lorazepam before each scan, he could not complete either scan due to anxiety and agitation.

Participant JHUFXS4 had already completed MRI scans of 30 and 60 min on separate days at another institution. For this prior investigation, a psychologist met with him online regularly for weeks before the scans to practice holding still despite the noise. He had never had a PET scan. His mother began practicing relaxation and holding still while listening to MRI sounds for weeks before the session at JHU. His mother and an investigator accompanied him into the MRI chamber throughout the MRI scan. His mother sat at the operator’s booth throughout the PET scan to praise him for holding still during the PET scan.

The non-displaceable binding potentials (BP_ND_s) [32] of [^18^F]FPEB by reference tissue graphical analysis (RTGA) [76,77] in each VOI of two men with FXS were below the BP_ND_s of five age-matched men with TD [25,30,55] (Figure 3). mGluR_5_ expression was lower in the men with FXS than the men with TD (Figure 3).

#### 3.1.3. IND and JHU

Combined (IND and JHU) estimates of mGluR_5_ were significantly reduced in all eight volumes of interest (anterior cingulate, caudate, occipital, parietal, posterior cingulate, putamen, temporal, and thalamus) in the men with FXS (*N* = 9) in contrast to the age-matched men with TD (Figure 4, Table 2). Although the axial resolution of the IND scans was approximately twice that of the JHU scans, the combined results are striking.

Furthermore, a two-way analysis of variance on an initial pooled dataset showed that both independent variables of institution (IND and JHU, *df* = 1, F = 34.3, *p* < 0.0001) as well as diagnosis (FXS and TD, *df* = 1, F = 38.7, *p* < 0.0001) had non-random effects on regional estimates of BP [80] (Table 2).

## 4. Discussion

We showed the feasibility and safety of administering MRI and PET in two independent pilot samples of men with FXS. We applied PET to quantitatively measure the density of mGluR_5_s in cortical and subcortical brain regions of these men with FXS following exposure to [^18^F]FPEB), which is a first study to our knowledge. We found that mGluR_5_ density was significantly reduced in the cingulate, cortex, striatum, and thalamus in men with FXS in contrast to age-matched men with TD. The tracer is a novel, specific mGluR_5_ ligand to measure the density and distribution of mGluR_5_s in the brains of humans, which constitutes an effective tool to confirm target engagement of NAMs for mGluR_5_s. The feasibility of this complex protocol requires a multidisciplinary effort that includes mock scanner training and practice sessions taught with behavioral psychology.

### 4.1. mGluR_5_s in Humans with FXS

#### 4.1.1. Feasibility of a Complex Protocol of MRI and PET Scans in Participants with FXS

##### Adults

A primary goal of this study was to determine the feasibility and safety of a complex protocol that included MRI and PET scans on men with FXS. We showed that this challenging protocol can be accomplished with mock scanner training and practice sessions taught with behavioral psychology [62,63,64] and trained parents. Additionally, an investigator and a parent routinely accompanied participants into the MRI chamber to assist with the process during the entire MRI series. Since state-of-the-art PET scanners provide three-dimensional image reconstruction, face masks may no longer be required to stabilize heads. Scans may be accomplished with gauze for optimal comfort.

We recommend several modifications to facilitate the completion of the MRI and PET scans on individuals with FXS. Mock scanner training beginning online for weeks before the actual scans provides the means to train participants and parents to relax quietly without moving while loud noises like a jackhammer are played [62,63,64]. Behavioral psychologists can meet with participants and parents repeatedly online to utilize training sessions for holding still while MRI soundtracks are played through recordings. The sessions can begin with short practices of 15 s. Gradually, the duration of the session can be increased to 30 or 60 min to train participants to calmly endure the challenges of the noise and stillness. Additionally, behavioral psychologists can provide the example of providing positive feedback to the participants. In other words, praising the participant for holding still during the practice session is a valuable positive reinforcement for desired behavior. On the other hand, criticizing the participant for moving may increase anxiety and lead to agitation and uncooperative behavior. Therefore, parents can be taught to reward the desired behavior.

Another approach to facilitate successful completion of scans includes the shortening of the duration of PET scans and the use of gauze instead of a rigid face mask. Additionally, performing PET and MRI scans on two separate days allows participants to recover from the stress of one scan before undergoing the next. The use of PET/MRI machines would simplify the protocol to accomplish both PET and MRI in a single session [81].

##### Adolescents and Children

Since PET involves greater than minimal risk due to radiation exposure, the safety and efficacy must be shown in adults before exposing vulnerable populations. For this reason, the current protocol was administered only to adults with FXS. After safety and efficacy are established in adults, then the procedure will be sequentially administered to adolescents, followed by children. The procedure may be modified for children to reduce the duration of scans. The procedure of the IND to conduct a 30 min scan 90 to 120 min after radiotracer injection with gauze to stabilize the head will shorten the stress of remaining on the scanner table. Another modification will be the utilization of PET/MRI scanners to conduct both PET and MRI scans in a single session instead of separate sessions for PET and MRI scans [81]. Mock scanner training by experienced behavioral psychologists [62,63,64] will be crucial to prepare children and adolescents for scans. Additionally, the participation of parents for each step is key to the accomplishment of this challenging protocol.

### 4.2. mGluR_5_ Measurement in Men with FXS

Another goal of this investigation was to find out if the study protocol can quantify mGluR_5_ expression in the brains of adult males with FXS. The data from our study show that the PET ligand binds mGluR_5_s in the brains of men with FXS, and that the expression of these receptors is decreased. This finding could be mediated by excessive upstream signaling resulting in reduced expression of mGluR_5_s. Internalization of the mGluR_5_s [20,82] throughout the brain induced by the radiotracer, the scanner, or other aspects of the environment of PET scans may explain the reductions in mGluR_5_s in our participants with FXS.

A preliminary attempt to perform an analysis on a combined dataset of both FXS/TD data from the IND and JHU showed that the source of the data was a non-random factor that influenced the outcome variable. We shall strive to reduce this possible confounding influence to improve the effect size in future analyses. As a future direction, we are developing other means of analyzing larger datasets from multiple institutions in a comparable manner so that the data can be pooled after removing the confounding factors of methodological differences.

### 4.3. Avoiding Effects of Diurnal Variations of mGluR_5_s

We administered PET scans to participants with FXS at the same time of day (1 PM) to minimize effects of diurnal variations of mGluR_5_s. Participants with TD received radiotracer injections 32 ± 120 min (range −135 to +163) from 1 PM [55], resulting in a confounding influence of diurnal variation. Large alterations in radiotracer uptake on the same individuals during the same day suggest that there may be considerable diurnal variation in mGluR_5_s, with increased uptake later in the day [21,83,84,85]. Participants with FXS may experience greater anxiety with scans than participants with TD. Anxiety may increase cortisol values and result in diurnal variations. Thus, we assume that our participants with FXS likely exhibited the maximal radiotracer uptake at the time of their scans. Measurement of cortisol levels and administration of PET scans at the same time of day to all participants minimizes the effects diurnal variations of mGluR_5_s.

### 4.4. Limitations and Future Studies

There is a need for comprehensive protocols uniformly administered to all cohorts. The use of different protocols for PET at the collaborating institutions [52,86] confounds comparisons and contrasts of the results. Future investigations at multiple centers will benefit from the use of identical protocols and analyses for PET and MRI conducted contemporaneously. Analysis of results by a single center will facilitate the uniformity of the findings. Despite different protocols, the uniformity of the finding of reduced mGluR_5_ expression in multiple brain regions independent of protocol strengthens this study’s key finding.

Administration of the full neuropsychological battery to contemporaneous cohorts at all participating centers will provide the foundation to apply statistical analyses. Normalization of cognitive test scores for participants with FXS will remove a “floor effect” [58]. Future studies will benefit from examining participants with FXS exhibiting a spectrum of ID and ASD and comparison groups without FXS with levels of ID matched to the participants with FXS.

The current pilot study is limited by the incomplete *FMR1* gene and epigenetic (methylation) parameter identification, and incomplete size mosaicism and quantification of FMRP. Future studies will be enhanced by including these measures and whole exome sequencing (WES) [87] on all participants to test the hypothesis that the parameters are correlated [12].

Since increased protein synthesis has been demonstrated in fibroblasts of individuals with FXS and *fmr1* KO mice [88], measurement of protein synthesis, particularly in the mTOR and ERK signaling cascades, would be a valuable parameter to correlate with mGluR_5_ density and distribution in future investigations of FXS in humans. However, the absence of increased protein synthesis in young men with FXS sedated with dexmedetomidine for PET with L[1-^11^C]leucine suggests that humans with FXS may not demonstrate the increased protein synthesis seen in animal models [89].

#### Multimodal Imaging

Multimodal imaging can enhance future investigations by linking PET, electroencephalography [90], event-related brain potential (ERP) [91], resting state functional magnetic resonance imaging (rs-fMRI), diffusion tensor imaging (DTI), and movement measurement [92], along with quantitative measurements of FMRP and *fmr1* [3]. Newly developed PET/MRI scanners [81] may produce visualization of the density and distribution of mGluR_5_s that is superior to images obtained from HRRT co-registered with MRI. PET/MRI units are appealing for future investigations because a single session would be required. PET/MRI provides both functional (PET) and structural (MRI) findings in one encounter. Thus, PET/MRI instrumentation and many other multimodal techniques may be utilized when available for subsequent investigations.

## 5. Conclusions

We showed the feasibility and safety of applying PET as a tool to quantify mGluR_5_ receptor expression in the brains of humans with FXS.

We showed that the proposed protocol of MR and PET scans in one day is feasible in individuals with FXS who have received mock scanner training by an experienced behavioral psychology team.

Most importantly, we found for the first time that quantified mGluR_5_ expression using [^18^F]FPEB is reduced in the living human brain of men with FXS in contrast to healthy normal age- and sex-matched controls with TD.

Larger studies with additional molecular biomarkers [93] are needed to expand on the feasibility finding of this protocol to evaluate the receptor expression of mGluR_5_s using [^18^F]FPEB as a helpful tool for the design of clinical trials of glutamatergic agents in FXS. 

## Figures and Tables

**Figure 1 brainsci-10-00899-f001:**
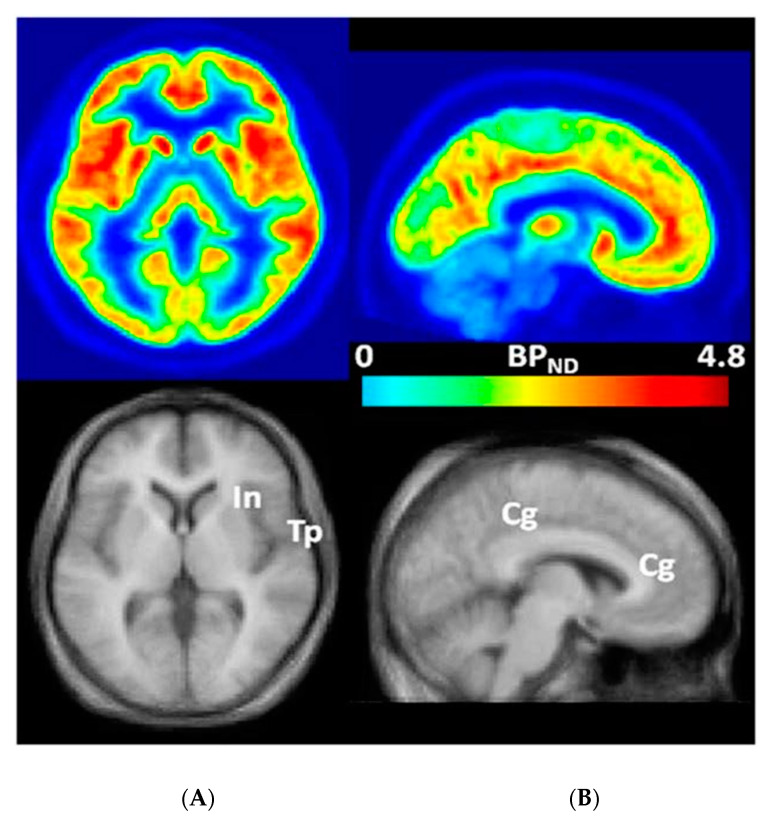
Transaxial (**A**) and sagittal (**B**) non-displaceable binding potential (BP_ND_) [32] images of [^18^F]FPEB (**top**) and matching magnetic resonance (MR) images (**bottom**) in statistical parametric mapping (SPM) [33] standard space. Regions with high BP_ND_ values, namely insular (In), temporal (Tp), and cingulate (Cg) cortices, are indicated on co-registered MR images [30]. This research was originally published in *JNM*. Wong DF, Waterhouse R, Kuwabara H, Kim J, Brašić JR, Chamroonrat W, Stabins M, Holt DP, Dannals RF, Hamill TG, Mozley PD. ^18^F-FPEB, a PET radiopharmaceutical for quantifying metabotropic glutamate 5 receptors: a first-in-human study of radiochemical safety, biokinetics, and radiation dosimetry. J Nucl Med. 2013;54:388-396. © SNMMI [30].

**Figure 2 brainsci-10-00899-f002:**
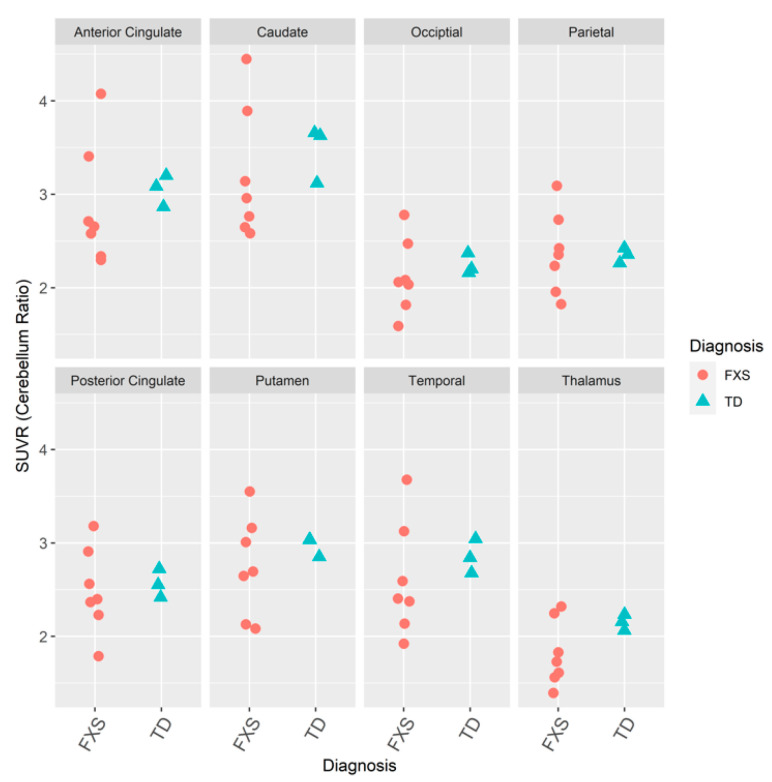
Dot plots of the ratio of densities of mGluR_5_ in volumes of interest (VOIs) to whole cerebellum for participants from the Institute for Neurodegenerative Disorders (IND) with FXS (*N* = 7) and TD (*N* = 3) who received intravenous bolus injections of 185 MBqs (5 mCis) of [^18^F]FPEB [55,79].

**Figure 3 brainsci-10-00899-f003:**
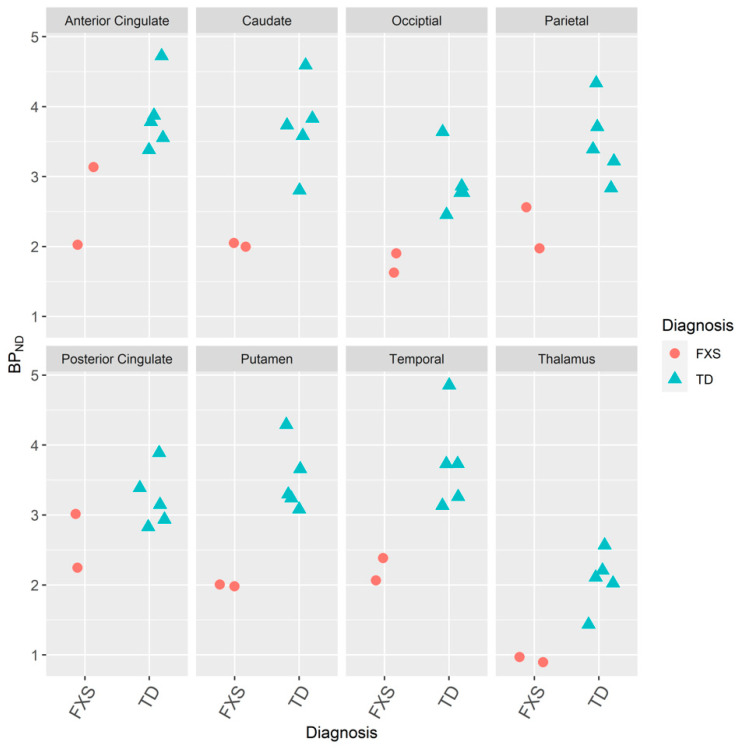
Dot plot of non-displaceable binding potential (BP_ND_) [32] images by reference tissue graphical analysis (RTGA) [76,77] of volumes of interest on positron emission tomography (PET) for 90 min of participants with FXS and ID (*N* = 2) and TD (*N* = 5) who received intravenous bolus injections of 185 MBqs (5 mCis) of [^18^F]FPEB [25,30,55,79].

**Figure 4 brainsci-10-00899-f004:**
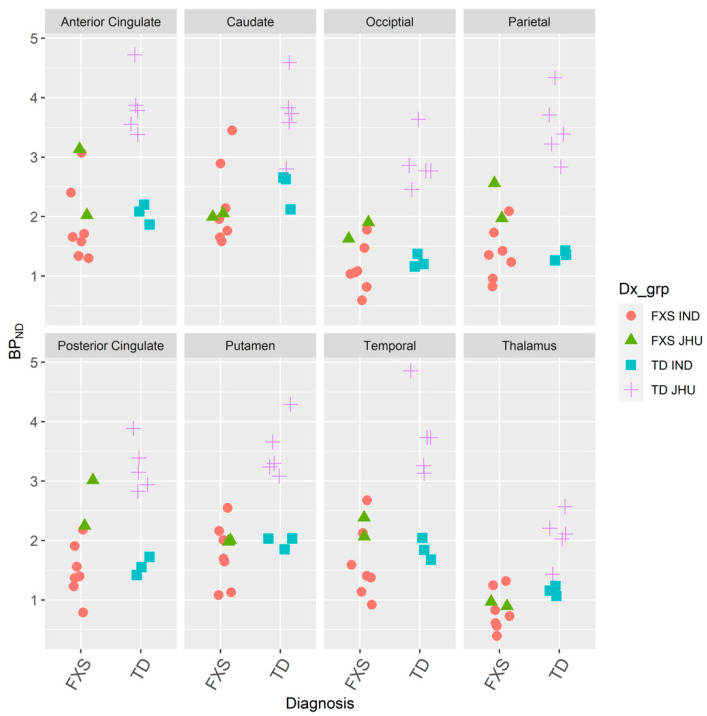
Dot plot of estimated binding potential (BP) [32] images by positron emission tomography after intravenous bolus injections of 185 MBqs (5 mCis) of [^18^F]FPEB [25,30] for men with fragile X syndrome (*N* = 9) and age-matched men with typical development (*N* = 8) from the IND and JHU [52,55,79].

**Table 1 brainsci-10-00899-t001:** Characterization of MRI sequences of the brain.

Format	Time to Repetition (TR) (ms)	Time to Echo (TE) (ms)	Thickness (mm)	Number of Slices
T1 sagittal	500	8	5.0	21
T1 SPGR recalled acquisition in the steady-state axial	35	6	1.5	124
T2 oblique	5900	95	5.0	27
DTI	12,100	88	2.0	72

Reproduced with permission [60]. The parameters of DTI include a slice thickness of 2 × 2 × 2 mm, field of view of 240 mm, iPAT (acceleration factor) of 2, 30 directions, and a b value of 1000 s/mm^2^. Abbreviations: MRI, magnetic resonance imaging; DTI, diffusion tensor imaging; SPGR, spoiled gradient.

**Table 2 brainsci-10-00899-t002:** Analysis of variance of estimates of mGluR_5_ expression by positron emission tomography after intravenous bolus injections of 185 MBqs (5 mCis) of [^18^F]FPEB in cortical and subcortical regions in the combined sample of men with fragile X syndrome (*N* = 9) and age-matched men with typical development (*N* = 8).

Region	Term	*df*	Sum of Squares	F Statistic	*p*-Value
Anterior Cingulate	Diagnosis	1	5.69	15.1	0.00165
	Source	1	5.93	15.7	0.00141
Caudate	Diagnosis	1	4.93	10.6	0.00569
	Source	1	1.22	2.62	0.128
Occipital	Diagnosis	1	4.37	23.1	0.000279
	Source	1	4.92	26	0.000163
Parietal	Diagnosis	1	5.32	18.9	0.000675
	Source	1	8.58	30.4	0.000076
Posterior Cingulate	Diagnosis	1	3.18	17.6	0.000906
	Source	1	7.04	38.9	2.18E-05
Putamen	Diagnosis	1	5.4	18.4	0.000753
	Source	1	3.1	10.6	0.00583
Temporal	Diagnosis	1	7.07	17.9	0.000834
	Source	1	5.92	15	0.00169
Thalamus	Diagnosis	1	3.32	23.7	0.000249
	Source	1	1.06	7.55	0.0157

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
