# Peer review of "Reduced Expression of Cerebral Metabotropic Glutamate Receptor Subtype 5 in Men with Fragile X Syndrome"

_brainsci, 2020, doi:10.3390/brainsci10120899_

Round 1

Reviewer 1 Report

Dear Authors,

your article entitled "Reduced Metabotropic glutamate receptor subtype 5 in men with fragile X syndrome" provides a pilot study focused on the expression of mGluR5 in vivo in a small cohort of mice and FXS individuals. Despite a great interest on this topic, I suggest some major revisions:

-greater conciseness (I encourage to shorten the MS) and more accuracy in the description (i.e. line 31 "relevant", line 45 "typical"; what is meant by relevant and typical?);

-statistical analysis is very lacking (in line 383 "significantly" has no a scientific value);

-discussion section should critically discuss and not describe the results;

-in line 385 ref. 112 should be changed with ref. 111;

-in the text references n.69-77 are not cited.

Reviewer 2 Report

Overall, a very interesting paper with novel techniques asking important questions to further our understanding in the field.  

Whilst, as ever, larger numbers (especially in the comparison groups), would improve the paper further, the results are still noteworthy and important as they stand.  

A few suggestions for minor revisions:
-In the procedures, it would be helpful if the authors could describe the MRI procedures in more detail. There are a few elements mentioned in different parts of the paper, but bringing together in the Methods section on procedures would be helpful.  The conclusions mentions that MR and PET were done on the same day - was this the case for all participants; and if not, how far apart were the scans.
-It would be helpful to other researchers in the field if the authors could comment on the feasibility/acceptability of the PET/MRI further and its impact on recruitment.  Were there any drop-outs during mock scanner rehearsals?  Any issues relating to the forehead gauze tape/fitted mask? Etc.  Any comment on relative acceptability of the tape vs mask approaches?  This would be helpful for other research teams considering PET studies.
-line 229 - could you clarify what ‘the following day’ relates to.  I presume this was the day following the neuro-cognitive assessments mentioned earlier, but would be helpful to clarify.  
-Figure 2 - although the reader can make their own inferences as to the distribution of results, showing the individual datapoints as well as the descriptives would give a fuller overview of the results.  Might also prompt discussion about variance in the FVBxC57 group.

Reviewer 3 Report

PET promises to be a tool to quantify mGluR5 receptor expression in the brains. The authors attempted to show that PET promises to be a tool to quantify mGluR5 receptor expression in the brains of fmr1 KO mice and humans with FXS.

However, the results were not congruent in the animal model vs humans. The authors should use the same radiotracer for both investigations of animals and humans which, as they mentioned, may show congruent results in the animal model and humans. Otherwise, there is no much relevance in publishing these incongruent findings together.

The authors aim to demonstrate that the proposed protocol of MR and PET scans in one day is feasible in individuals with FXS and ASD who have received mock scanner training by experienced behavioral psychologists.

However, the details of failure or success attempts were not documented (percentage of failures, details about the protocol, etc). The study did not include children, it was not mentioned why children were not included and the relevance of age in the proposed protocol of MR and PET scans.

Functional MRI and other imaging technologies can be altered by the psychological status of the patients, perhaps cortisol levels can alter diurnal levels, as well as, levels under anxiety provocation circumstances. FXS adults have higher levels of anxiety and while they may be able to cooperate for the scan, the anxiety of these participants can be significantly higher than the anxiety of the controls. The authors should mention this in the discussion.

Most importantly, the authors aim to show that quantified mGluR5 expression using [18F] in humans with FXS is significantly reduced in the living human brain of men with FXS in contrast to controls such as men with FXS-M, men with ASD, and individuals of both genders with TD. The authors argued a significant reduced mGluR5 expression in the living human brain of men with FXS in contrast to controls.

However, the analysis was performed separately for the 2 institutions and it is unspecific. This is a very important finding for the field and the authors should pursue this objective under a precise project and analysis.

The IND (Institute of neurodevelopmental or neurogenerative disorders? - the term is not consistent in the manuscript) included 7 FXS compare with a FXS-M and a TD female. This comparison is not appropriate, the FXS-M cannot be included as a control/TD; the other control is a TD female, sex differences may exist. The authors should compare the 7 men with the FXS and 7 TD men.

The JHU conducted an analysis of 2 FXS/ASD men, 6 ASD, and 3TD. This comparison is also inappropriate why is there an ASD group here? Was there a through genetic testing in the ASD group, such as WES? What happened to the other 2 FXS/ASD males? If the purpose is to describe FXS why the number of ASD participants is higher than the FXS patients, 200%. If the rates of ASD in FXS are 40-60% then the authors are studying a subgroup of FXS patients that may not represent the majority of the FXS group. It is not clear why all the FXS patients had ASD diagnosis, did they exclude no ASD-FXS patients. If they did the focus is ASD and they should focus on specific profiles of ASD in both groups, otherwise is a highly clinically heterogeneous group in which small numbers are very problematic.

It may be better to compare 2 FXS/regardless of ASD and 2 TD males or a symptom-based approach for autism in the ASD group, the FXS/ASD group and a TD group with all matched for sex and age.

Round 2

Reviewer 1 Report

The MS in the present form may be accepted for publication in Brain Sciences.

This manuscript is a resubmission of an earlier submission. The following is a list of the peer review reports and author responses from that submission.